# Secret-Key Agreement by Asynchronous EEG over Authenticated Public Channels

**DOI:** 10.3390/e23101327

**Published:** 2021-10-11

**Authors:** Meiran Galis, Milan Milosavljević, Aleksandar Jevremović, Zoran Banjac, Aleksej Makarov, Jelica Radomirović

**Affiliations:** 1Vlatacom Institute of High Technology, Milutina Milankovica 5, 11070 Belgrade, Serbia; meiran.galis@gmail.com (M.G.); zoran.banjac@vlatacom.com (Z.B.); aleksej@vlatacom.com (A.M.); jelica.radomirovic@vlatacom.com (J.R.); 2Technical Faculty, Singidunum University, Danijelova 32, 11000 Belgrade, Serbia; ajevremovic@singidunum.ac.rs

**Keywords:** key distillation, advantage distillation, information reconciliation, CASCADE, EEG, Wisconsin Card Sorting Test

## Abstract

In this paper, we propose a new system for a sequential secret key agreement based on 6 performance metrics derived from asynchronously recorded EEG signals using an EMOTIV EPOC+ wireless EEG headset. Based on an extensive experiment in which 76 participants were engaged in one chosen mental task, the system was optimized and rigorously evaluated. The system was shown to reach a key agreement rate of 100%, a key extraction rate of 9%, with a leakage rate of 0.0003, and a mean block entropy per key bit of 0.9994. All generated keys passed the NIST randomness test. The system performance was almost independent of the EEG signals available to the eavesdropper who had full access to the public channel.

## 1. Introduction

The information-theoretic approach to information security has received renewed attention due to recent advances in quantum computing. The central principle of this approach is simple to formulate: a cryptographic system provides absolute secrecy (information-theoretical secrecy) of messages, if, and only if, the uncertainty (entropy) of its secret key is not less than the uncertainty of messages [1]. Systems designed in this way are known to be resistant to the unlimited computing resources of adversaries, and thus to cryptanalysis based on the use of quantum computers [2].

Therefore, it can be said that we have entered an age in which the “harvest” of the uncertainty of every possible type, origin, and place of collection, becomes a priority task for generating and distributing cryptographic keys with maximum entropy.

In this context, the fundamental results of Ahlsvede and Csiszar [3], Maurer [4], and Csiszar and Narayan [5] deserve special attention. The basic idea of this approach consists in extracting mutually correlated signals of sufficiently large entropies.

The following two approaches can be distinguished, based on the location of the source of uncertainty, ref. [4]:

(i) extraction of signals from sources that are independent of communication channels (the *source model*), and

(ii) extraction of signals from used communication channels (the *channel model*).

In this study, we explore the possibility of extracting cryptographic keys from electroencephalography (EEG) signals, applying a source-model-based approach. In our case, the EEG signals were recorded using the 14-channel EMOTIV EPOC+ wireless EEG headset [6,7]. The choice of EEG as a source of randomness was motivated by two factors.

First, the role of the secret-key agreement (SKA) is to ensure the establishment of symmetric encryption keys for participants who do not possess previously distributed identical secret keys. This is typical in some military applications, in which keys cannot be established through physical distribution, or in scenarios of secret and special operations, in which participants do not have pre-generated and distributed secret keys. The separation of functional blocks is a basic principle in the design of professional information security systems because it minimizes the risk of compromising the entire system by compromising one part of it. Accordingly, an SKA system should be independent of cryptographic and telecommunications modules, which excludes the use of the SKA channel model. With only the SKA source model remaining, using the participant’s biometric signal would be of great advantage, eliminating the need for an additional random source, as well as the risks and costs associated with it (development, production, quality control, safe storage, etc.).

Secondly, when choosing the biometric signal, it is necessary to consider the commercial availability, robustness, and functionality of the corresponding sensor system. Among the candidate biometric signals, which include gait, motion, electromyography (EMG), electrocardiogram (ECG), and EEG (see review [8]), EEG stands out for its high entropy content, as well as for the commercial availability of EEG sensors of the required quality and robustness. The availability of EEG sensors and processing systems is primarily driven by their main role in modern human-computer-interface systems (see review [9]). In this regard, the EMOTIV EPOC+ system meets all our criteria. 

In Section 2, we provide arguments as to why a set of subjects exposed to a certain mental task can be considered an example of a single discrete memoryless source (DMS). Our research was conducted on the EEG signals of 76 participants recorded asynchronously while solving the Wisconsin Card Sorting Test (WCST) [10,11]. The WCST test has been chosen arbitrarily and can be replaced by any other task, such as reading a selected text or viewing a selected image [12].

In Section 3, we analyze the statistical and information-theoretic characteristics of this information source and identify the most important parameters of each phase of the proposed SKA, namely: advantage distillation (AD), information reconciliation (IR), and privacy amplification (PA).

Section 4 presents the results of an extensive experiment to obtain secret keys for all pairs of participants (76 × 75/2 = 2 = 2850 keys), for three types of an eavesdropper (referred to as Eve): Super evil Eve, Medium evil Eve, and Uninformed Eve. These types cover the entire range of prior information available to Eve. Section 5 continues by comparing the obtained results with the performance of related systems described in the available literature. Section 6 presents the security aspects of the proposed SKA, and the scenarios of practical application. Finally, in Section 7, we analyze possible approaches for increasing the secret key rate.

In the Conclusion, we discuss a number of open issues and point out a class of algorithms for generating and distributing secret keys based on the so-called data exchange problem [13,14]. Combining this approach with the SKA system presented in this paper will be the subject of our future research.

## 2. Virtual DMS Channel Based on a Chosen Mental Task

### 2.1. Sequential Key-Distillation Strategy

Figure 1 shows a source model for SKA within a scenario in which three parties, Alice, Bob, and Eve, observe realizations of a DMS. Each of them receives their own set of observations. Let X, Y, and Z, be Alice’s, Bob’s, and Eve’s observations, respectively. It is assumed that DMS is beyond the three parties’ control, though its statistics may be known to all of them. Alice’s and Bob’s goals are to agree on a secret key K, based on their observations X and Y, so that Eve has no information about it. In the SKA scenario, a public communication channel, through which Alice and Bob can exchange information, is fully available to all parties, including Eve. It is an underlying assumption that this public channel is authenticated so that no impersonation is possible.

The rules, under which Alice and Bob calculate the messages exchanged over a public channel and finally agree on the secret key, define a four-stage sequential key-distillation (SKD) strategy [4]:

*Randomness sharing*. Alice, Bob, and Eve observe *n* realizations of DMS (XYZ, P_XYZ_), where P_XYZ_ denotes the joint probability measure of the random variables X, Y, and Z.

*Advantage distillation*. If necessary, Alice and Bob exchange messages over a public channel to process their observations and to “distill” the observation parts on which they have an advantage over Eve.

*Information reconciliation*. Alice and Bob exchange messages over the public channel to process their observations and agree on a common binary string.

*Privacy amplification*. Alice and Bob publicly agree on a deterministic function that they would apply to their common sequence to generate the secret key. 

The secrecy capacity of a public channel is the maximum rate at which information can be reliably exchanged between legitimate parties such that the rate at which an eavesdropper obtains this information is arbitrarily small. The secret key capacity is thus the maximum length of a secret key that can be sent in the presence of an eavesdropper and can be defined by
C_k_ = min{I(X;Y), I(X:Y|Z)},(1)
where I(X;Y) denotes the mutual information between X and Y, while I(X;Y|Z) denotes this mutual information conditioned by Z. In the special case, when Eve is independent of Alice and Bob, i.e., when Z is independent of X and Y, the secret key capacity is equal to
C_k max_ = I(X;Y).(2)

The advantage of the SKD strategy is the proven achievement of all secret key rates lower than the secrecy capacity C_k_, as well as its explicit practical implementation [4].

Based on this strong theoretical result, we propose the application of an SKD strategy to generate random sequences from DMS (XY, P_XY_), where observations X and Y represent six-dimensional performance metrics signals obtained from the EMOTIV EPOC+ EEG headsets, worn by two subjects, asynchronously engaged in the same mental task (see Figure 2); P_XY_ denotes the joint probability measure of random variables X and Y. 

By comparing Figure 1 and Figure 2, it can be seen that the “common random source” in Figure 1 is replaced by the engagement in the same mental task. Unlike the classical setting of Figure 1, in which the correlation of the observed data is caused by the underlying physical phenomena, the observations (X, Y, Z) in Figure 2 are correlated due to the similar thought processes of the test participants. This correlation structure is invariant to:the time and place of the test, andthe tested subjects,
allowing for the asynchronous acquisition of EEG signals. This property is of particular importance in practical situations where synchronization is difficult to achieve or would require additional SKD system complexity and/or resources. 

### 2.2. An Experimental Environment for Recording EEG Signals of the Test Participants

For this work, the data were collected during sessions where participants were using different computer applications, including the Wisconsin Card Sorting Test. The sensors used during the sessions included electroencephalography and eye-tracking devices. The mouse movements and keyboard strokes were also recorded. The human-computer interaction monitoring and analytics platform (HCI-MAP) [15], whose architecture is presented in Figure 3, was used for the collection and synchronization of data (signals, application events, screenshots, etc.).

The electroencephalography signals were collected by the EMOTIV EPOC+ device, a wireless EEG headset with 14 channels designed for measuring the brain’s cortical activity [16]. The device uses A/D conversion with sequential sampling at a 128 Hz sampling rate. Its output frequency band was flat from 0.2 to 45 Hz and was digitally notched at 50 Hz and 60 Hz to remove interference from the electrical power supply. The device was connected to the HCI.MAP platform using a standard 2.4 GHz Wi-Fi connection.

The sessions were recorded within a study that involved 76 participants, aged 15–25 years, selected by a random sampling method, see Figure 4. The participants were aware of the research procedure, including the application of the sensors, and had voluntarily agreed to take the test. In addition, they were aware that the test would be conducted anonymously: their records were mixed and stripped of identifying information. The only personal data stored were gender, age, and educational level. The institutional ethics committee approved this research following the principles of the Declaration of Helsinki. The subjects were given a computerized version of the tests. The computer mouse and keyboard served as additional sensors. The medical criterion for inclusion in the study was the absence of neurological and psychiatric disorders, including addiction.

### 2.3. Acquisition of EEG Signals

As a result of real-time EEG measurement, 6-dimensional time series were obtained for each test participant, with each variable measuring a different so-called performance metric [7,17].

It is very important to note that these 6 metrics for the proposed SKA system represent a six-dimensional source of common randomness. It was derived based on 6 fixed transformations, which were consistently applied in the same way to the 14th channel EEG of each participant. Therefore, any neuropsychological or neurophysiological interpretation of these metrics and the question of their reliable connection with the mental states of the test participants are irrelevant for our system.

Figure 5 shows the recorded signals of Alice, Bob, and Eve, randomly selected among all 76 test participants.

The signal acquisition was followed by the dimensionality transformation from 6 to 1 dimensions, resulting in a univariate time series for each participant. As it is important to preserve the correlation structure between the participants, the applied transformation consisted of simple serialization. Namely, at each sampling point, a buffer accepted a 6-dimensional measurement vector, and then sent its components out serially. The resulting one-dimensional signals for Alice, Bob, and Eve from Figure 5 are shown in Figure 6.

From Figure 6, one can see that the pre-processing transformation described above had indeed preserved the inherent correlation structure of the subjects’ performance metrics signals. Henceforth, this preprocessed signal will be referred to as the “primary EEG source”.

The next preprocessing step involved quantization of the primary EEG source. This problem has been frequently examined in the literature, but primarily concerning the sequential key distillation strategies for the channel model [18,19,20]. A substantial difference between the discrete and continuous sources was shown in [21,22] (see Remark 5 in [21]). For discrete sources, when the data rate over a public channel is greater than H(X|Y), the upper limit of the secret key extraction rate can be achieved even without quantization, by applying Slepian–Wolf coding and the privacy amplification (PA) procedure [23]. On the other hand, for continuous Gaussian sources, the upper limit cannot be reached for any finite data rate over a public channel. In [24] (Proposition 5.6), it was shown that if Xq is a uniformly, finely-enough-quantized version of X, mutual information I(Xq;Y) approaches the original I(X;Y) exponentially fast with the increase of data rate on a public channel. Therefore, sophisticated quantization schemes, e.g., TCVQ (trellis coded vector quantization scheme), make sense only in conditions of limited communication over a public channel. Since the primary goal of this work was an experimental confirmation of the proposed concept, without the public channel data rate limitation, we opted for the simplest scalar uniform quantization.

The Shannon, or block entropy [25], is given by
(3)$$H_{n}=-\underset{a_{1}{,a}_{2}{,\ldots ,a}_{n}}{\sum }{P(a}_{1}{,a}_{2}{,\ldots ,a}_{n}{)\log}_{2}{P(a}_{1}{,a}_{2}{,\ldots ,a}_{n})$$
where ${P(a}_{1}{,a}_{2}{,\ldots ,a}_{n})$ is the probability of occurrence of the pattern $a_{1}{,a}_{2}{,\ldots ,a}_{n}$ in the output of an information source. This entropy is known as the n-block entropy. The normalized block entropy refers to the quantity $\frac{H_{n}}{n}$, whose asymptotic value $\underset{n\to \infty }{\lim}\frac{H_{n}}{n}$ is known as the Shannon or block entropy rate. In practice, we are interested in the entropy of finite sequence x of length N. If one regards finite sequence x as representative output from some information source, one may estimate $P\left(a_{1}{,a}_{2}{,\ldots ,a}_{n}\right)$ from the pattern frequencies observed in x. If x is a binary sequence, the pattern frequencies are equal to the set of all binary n-grams, while normalized block entropy is equal to normalization to one bit of x. 

Figure 7 shows the change in normalized block entropy of the analyzed primary EEG source, as a function of the number of bits per sample quantized by a uniform quantizer. This function was calculated for the values of the block length change from 1 to 20. 

From Figure 7, one can see that an increase in the number of bits per sample (i.e., word length or bit depth) lead to an initial increase and then a decrease in normalized block entropy. An increase of the block entropy in the range n_b_ = [1,7], where n_b_ is the number of bits per sample, corresponds to the better description of the information content of the primary EEG source. The subsequent decay in normalized block entropy in the range n_b_ = [8,16] can be interpreted as over-quantization, which introduces additional redundancy in the primary EEG source. Many authors have noticed, see for example [26], that over-quantization may increase the secret key extraction rate. With this phenomenon in mind, we decided to design a system operating with two different quantization values: n_b_ = 5, which corresponds to the under-quantization mode, and n_b_ = 10 for the over-quantization mode, and to investigate their impact on the overall system performance. 

## 3. System for Sequential Secret Key Agreement Based on the Primary EEG Source

### 3.1. Statistical and Information-Theoretic Characteristics of the Primary Source

Figure 8 and Figure 9 show the basic characteristics of the primary EEG sources for n_b_ = 5 and n_b_ = 10, respectively. The basic characteristics include the histogram of the signal sequence length for all 76 test participants and the histogram of normalized Hamming distances D_h_ of all their pairs. The normalized Hamming distance between two binary sequences X and Y of the same length is given by the expression
(4)$$D_{h}(X,Y)=\frac{\mathrm{number}\;\mathrm{of}\;\mathrm{non}-\mathrm{match}\;\mathrm{bits}}{\mathrm{number}\;\mathrm{of}\;\mathrm{bits}\;\mathrm{compared}}$$

For sequences of different lengths D_h_ is calculated based on the expression:D_h_ (x_1_,x_2,_…,x_n_,y_1_,y_2_,…,y_m_) = D_h_ (x_1_,x_2_,…,x_p_,y_1_,y_2_,…,y_p_), p = min(n,m).(5)

In this way, we minimized the rejection of available data during the SKD operation and evaluation of its performance on all of the pairs of participants. Assuming that the primary EEG source sequences consist of binary iid random variables, the conditional entropy H(X|Y) and the mutual information I(X,Y) become:H(X|Y) = h_b_(D_h_(X,Y)), I(X,Y) = H(X) − h_b_(D_h_(X,Y))(6)
where h_b_ is the binary entropy function,
(7)$$h_{b}(p)=-{\mathrm{plog}}_{2}(p)-(1-p)\log_{2}(1-p),\;p\in \left[0,1\right].$$

Given that the function h_b_ is monotonically increasing in the range [0, 1/2], D_h_ (X, Y) measures well the maximum extraction rate of secret keys Ck, given by Equation (1), for a fixed amount of information that Eve has about sequences X and Y.

Recall that in the case of random and completely independent sequences, the histogram of their mutual normalized Hamming distances is narrowly centered around the value of 0.5. To be convinced of this statement let $S_{i}$ be the binary random variable denoting whether two binary strings X and Y of length p, differ in position i. These p random variables are independent, with equal probability of 0 and 1, i.e., Prob{Si=0}=Prob{Si=1}=12. By linearity of mathematical expectation,
$$E\left\{S_{1}{+S}_{2}{+\ldots +S}_{p}\right\}=E\left\{S_{1}\right\}+E\left\{S_{2}\right\}+\ldots +E\left\{S_{p}\right\}=$$ 1/2 + 1/2 + … + 1/2 = p/2. Consequently,
(8)$$E\left\{D_{h}\left(X,Y\right)\right\}=\frac{1}{p}E\left\{S_{1}{+S}_{2}{+\ldots +S}_{p}\right\}=\frac{1}{p}\cdot \frac{p}{2}=\frac{1}{2}.$$

By comparing the right-side histograms in Figure 8 and Figure 9, a shift towards smaller normalized Hamming distances (i.e., smaller differences between the signals) can be observed. This again shows that over-quantization introduces additional correlation to the ensemble of realizations of the primary source. As the sampling rate for the EMOTIV EPOC+ device was 2 samples per second, it follows that the test duration ranged from 83 to 500 s, with a mean value of approximately 250 s.

### 3.2. Eavesdropper Model

Since the experimental evaluation of the system was performed on a group of participants, we can distinguish three typical scenarios from the point of view of an eavesdropper (Eve), according to the degree of available prior information about the primary source.

The eavesdropper is an insider who not only knows who Alice and Bob are but also has all the EEG signals of the test participants, except the signals of Alice and Bob. Additionally, the attacker on the system knows which of the participants’ signals is closest (most similar) to the signals of Alice and Bob. So, for each pair (Alice, Bob), the attacker can adaptively choose Eve, who is the closest to Alice and Bob in terms of the normalized Hamming distance. This, theoretically and practically, imposes the most difficult conditions for extracting secret keys, about which Eve should not have any information. Therefore, this type of eavesdropper is colloquially named “Super evil Eve”-SE.The eavesdropper does not know who Alice and Bob are, so he chooses Eve whose position is equally distant from all participants in terms of the normalized Hamming distance, which is equivalent to a centroid of a cluster that encompasses the entire population. Therefore, we colloquially called this Eve the “Medium evil Eve”-ME. For the analyzed primary source, ME corresponds to subject N^o^ 62, see Figure 10.The eavesdropper has no specific information about the primary source, except that it consists of EEG signals obtained by the EMOTIV EPOC+ device. The optimal strategy for the attacker, in this case, is to record his EEG signal and participate in the protocol with it as Eve. We colloquially called this Eve “Uninformed Eve”-UE. In the conducted experiments, UE is a subject outside the group of test subjects, whose EEG was recorded during the observation of one image, more precisely the reproduction of the famous icon, the “White Angel”, from the Serbian medieval monastery, Mileševa [12,27] for 768 s. Within the conducted cluster analysis, this subject is referred to by numeral 76, see Figure 10.

Figure 10 shows the dendrogram for hierarchical cluster analysis of the primary source signal, constructed by Ward’s method [28]. The input to the clustering procedure is a matrix, formed by the normalized Hamming distances. One can note that the subject UE does not differ significantly from other test participants, although his EEG signals resulted from a completely different mental task. 

### 3.3. Structure of the Proposed SKD System

Figure 11 shows the basic block structure of the proposed SKD system. The ultimate goal of the system is to ensure that the final secret keys of the legitimate participants in the protocol (Alice and Bob) be identical, *K_A_ = K_B_*, while Eve’s key *K_E_* should not carry any information about them. Following the basic Kerckhoffs principle-security is not obscurity [29], Eve knows all the elements of the system and all the parameters of individual subblocks. In [4] it is shown that the optimal strategy for Eve is to repeat the same actions that Alice and Bob agree on over a public channel. Serialization and uniform quantization are followed by advantage distillation, information reconciliation, and privacy amplification. PA is realized by applying a selected family of universal hash functions. At the end of the system operation cycle, Alice and Bob share an identical secret key, *K_A_ = K_B_*, while Eve’s key, *K_E_*, does not carry any significant information about them.

#### 3.3.1. Advantage Distillation (AD)

In the general case, it is necessary to assume that Eve has an initial advantage over Alice and Bob, i.e., that the normalized Hamming distance between Eve’s string and the one of Alice (or Bob) is less than the normalized Hamming’s distance between Alice’s and Bob’s strings. The goal of the advantage distillation (AD) phase is for Alice and Bob to exchange messages over a public channel, which will result in reversing the advantage in their favor. 

Several AD algorithms have been reported in the literature. The most widely known among them are the bit-pair (BP AD) protocol [30] and the more recent bit-pair advantage distillation/degeneration protocol (BP ADD) [31]. The main difference between these two protocols is that, unlike BP AD, BP ADD not only reduces the normalized Hamming distance between Alice’s and Bob’s sequences but also increases the distance between Eve’s and Alice’s (Bob’s) strings [32].

Below is a description of the AD (Algorithm 1) and ADD (Algorithm 2) protocols. Xi and Yi denote the *i*-th bit of sequences initially owned by Alice and Bob, respectively.

**Algorithm 1** Bit Parity AD protocol
1:Alice and Bob group $n_{\mathrm{AD}0}$ bits into 2-bit blocks.2:Alice and Bob compute the parity bits of these blocks, $\left\{X_{2i+1}⨁X_{2i+2}\;|\;i=0,1,\ldots ,\lfloor \frac{n_{\mathrm{AD}0}}{2}\rfloor -1\right\}$,3:Alice sends $\left\lfloor \frac{n_{\mathrm{AD}0}}{2}\right\rfloor$ parity bits to Bob over the public channel. If the parities match, Bob announces OK on the public channel.4:Both Alice and Bob keep the first bits of these selected 2-bit blocks to form a new, shorter string, which serves as the input bit string for the (s+1)th round


**Algorithm 2** Bit Parity ADD protocol
1:For k=1,2,…, Alice computes $C_{k}{=X}_{2k-1}⨁X_{2k}$ and sends $C_{k}$ to Bob;Bob computes $D_{k}{=Y}_{2k-1}⨁Y_{2k}$ and sends $D_{k}$ to Alice.2:If $C_{k}\neq D_{k}$, Alice deletes $X_{2k-1}X_{2k}$ from X and Bob deletes $Y_{2k-1}Y_{2k}$ from Y.If $C_{k}{=D}_{k}$, Alice judges whether $X_{2k}=1$ holds or not; if $X_{2k}=1$, Alice deletes $X_{2k-1}$ from X, otherwise, Alice deletes $X_{2k}$ from X. Similarly, Bob judges whether $Y_{2k}=1$ holds; if $Y_{2k}=1$, Bob deletes $Y_{2k-1}$ from Y, otherwise, Bob deletes $Y_{2k}$ from Y.


The efficiency of the BP AD and the BP ADD protocols can be assessed from Figure 12, Figure 13, Figure 14 and Figure 15, which show the evolution of the distribution of the corresponding normalized Hamming distances during the first two iterations of these protocols. Iteration 0 (colored in blue) denotes the initial distribution of the normalized Hamming distances of the available primary source sequences. By comparing the mean values of these distributions at the end of the second iteration (green), for the BP AD protocol (Figure 12 and Figure 13) and the BP ADD protocol (Figure 14 and Figure 15), it is readily seen that Alice and Bob achieved a significant advantage over Eve with the BP ADD protocol. It will be further confirmed through a complete experimental evaluation.

#### 3.3.2. Information Reconciliation (IR)

After the AD phase, Alice knows much more about Bob’s sequence than Eve. The goal of the IR phase is for Alice to get the full and exact knowledge of Bob’s sequence. All protocols of this class use an iterative procedure for detecting and correcting errors (discrepancies) in Alice’s and Bob’s sequence, based on two-way communication over a public channel. After the detection and correction of all errors, Alice’s and Bob’s sequences exactly coincide, fulfilling the objective of this phase. Although a whole family of IR protocols has been developed based on powerful error-correcting codes, such as low-density parity-check codes [33], here we opted for one of the most widely used and most efficient IR algorithms, the so-called Cascade protocol (Algorithm 3), first proposed in [34]. It is generally believed that this protocol provides significantly less information to Eve about the common sequence extracted by Alice and Bob, compared to complex error-correcting algorithms. This protocol has found wide application in the domain of quantum key distribution, and, as such, has been continuously improved and optimized. In this paper, we used an implementation described in [35] and its associated GitHub repository [36].

The Cascade information reconciliation protocol proceeds in several iterations. Alice’s and Bob’s sequences are divided into blocks in each round and the parity of their blocks is compared, which allows the finding and correction of an error in the event it occurs. The number of iterations and the block size of the first iteration is determined by Alice and Bob before execution.

**Algorithm 3** Cascade protocolInput: A, B %keys of Alice and BobOutput: K %reconciled key
1:In the first iteration, Alice and Bob divide their strings into blocks and Alice sends the parities of all her blocks to Bob2:Bob calculates his parities and proceeds with the binary algorithm (Algorithm 4)3:At the beginning of every other iteration, Bob needs to reshuffle bits of his key and repeat steps 1 and 2, but using larger blocks, new block size=2 old block size4:Corrected bits will cause a cascade effect on the shuffled blocks from earlier iterations, so we go back and apply the binary algorithm to those blocks5:Repeat steps 3 and 4 until the number of iterations is reached


**Algorithm 4** Binary algorithmWhen blocks of keys A and B have different parity:
1:Alice divides A into two halves and sends Bob the parity of the first half of A2:Bob divides B in the same way and compares the parity with Alice’s to determine which half contains an odd number of errors3:Apply these two steps repeatedly until an error is found.


Examples 1, 2, and 3 illustrate the operation of the AD, ADD, and Cascade protocols, respectively.
**Example 1** Bit Parity AD protocolInput: A = 1101000111011001001000100111111101    B = 1100001000111010101000010110010010
1.  A = 11|01|00|01|11|01|10|01|00|10|00|10|01|11|11|11|01
    B = 11|00|00|10|00|11|10|10|10|10|00|01|01|10|01|00|10
2,3: A = 11|01|00|01|11|01|10|01|00|10|00|10|01|11|11|11|01
    B = 11|00|00|10|00|11|10|10|10|10|00|01|01|10|01|00|10
4.   A = 100110101010
    B = 101011100001

**Example 2** Bit Parity ADD protocolInput: A = 0100000011010100101101001100101100    B = 1010001100110111011000000000000011
1:   A = 01|00|00|00|11|01|01|00|10|11|01|00|11|00|10|11|00
   B = 10|10|00|11|00|11|01|11|01|10|00|00|00|00|00|00|11
2:   A = 01|00|00|00|11|01|01|00|10|11|01|00|11|00|10|11|00
   B = 10|10|00|11|00|11|01|11|01|10|00|00|00|00|00|00|11
3:   A = 100110101010
   B = 101011100001

**Example 3** Cascade protocol

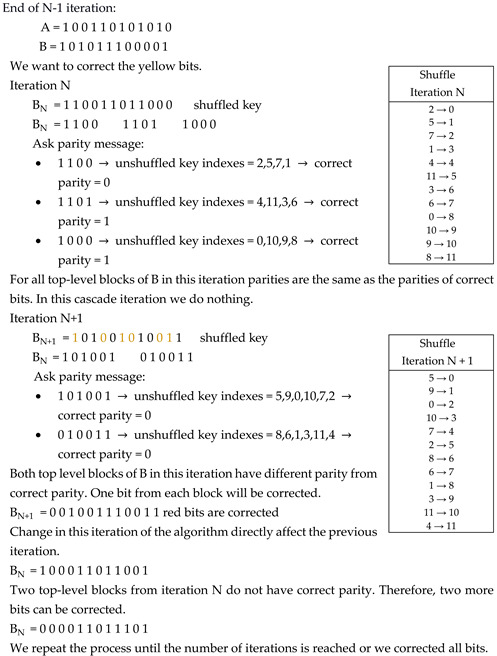



#### 3.3.3. Privacy Amplification–(PA)

During the execution of any IR protocol, observation of the public channel provides some partial knowledge to Eve about the common sequence extracted by Alice and Bob. Therefore, the last step in the SKD strategy is the application of an appropriate transformation, which will reduce Eve’s information to a negligible amount. Suppose that, during the Cascade protocol execution, Eve received a set of information about the parity of the individual blocks of the final common sequence. From the point of view of cryptanalysis, this is equivalent to a kind of algebraic attack, in which the attacker can compose a set of linear equations (corresponding to each parity query) over the unknown bit values of the common sequence. The dominant approach in the design of PA algorithms is based on the well-known leftover hash lemma [37]. It provides an answer to the question of whether a cryptographic key of length *n*, about which the opponent knows the values of some *t* bits (*t < n*), can still be used or must be discarded in favor of a new key. The answer is that we can use such a partially compromised key and, by appropriate transformation, produce a key of length around *n-t* bits, about which the opponent knows almost nothing. In [37] it was shown that the mentioned transformation can be any hash function $g:\;{\left\{0,1\right\}}^{n\;}\to \;{\left\{0,1\right\}}^{k}$ from the so-called universal class of hash functions (here *k* is the length of the output hash string). In the experimental evaluation of the proposed SKD system, we used a universal class of hash functions, given by
(9)$$H=\left\{h_{M}:\;M\in {\mathrm{GF}(2)}^{k\times n}\right\},$$
(10)$$h_{M\;}=\mathrm{Mx}$$
where *M* denotes a binary matrix of dimensions *k × n*, while all operations are performed in a two-element Galois field, *GF*(2). If *n_p_* is the number of parity queries exchanged over the public channel during the execution of the Cascade protocol, then the inequality *n_p_ > t* holds, where *t* is the number of bits of the Alice–Bob common sequence that Eve can know after completion of the IR phase. In the worst-case scenario, Eve gains knowledge of one bit of the Alice–Bob common sequence for each new parity query. In this case, t = n_p_, and therefore dimension k of matrix M in Equations (9) and (10) becomes:(11)$${k=n-n}_{p}.$$

Since the starting key length and the number of parity queries are known (values *n* and n_p_), k is also known, so the hash functions given by Equations (9) and (10) can be calculated and applied, giving a final common Alice–Bob secret key. According to the leftover hash lemma, this results in Eve’s key *K_E_* carrying negligible information about the established common final secret key *K_A_ = K_B_* between the legitimate parties, Alice and Bob.

## 4. Results 

The proposed SKD system was tested on two types of primary EEG sources, obtained for two quantization values: *n_b_* = 10 and *n_b_* = 5 bits per sample. In the context of advantage distillation, two variants of SKD were tested, the first with the BP AD algorithm and the second with the BP ADD algorithm, henceforth abbreviated as AD and ADD, respectively. The tests were performed on all 76 × 75/2 = 2850 pairs of subjects, for two quantization variants and all three types of Eve (EE, ME, UE). The number of the AD algorithm iterations was set at *n_a_* = 2. It has been shown in practice that this number of iterations was sufficient to achieve a significant advantage for Alice and Bob over all Eve types. The selected value of the parameter *n_a_* is a trade-off between maximizing the advantage over Eve and minimizing the resulting loss of sequence length at the output of the AD stage. 

In all quantization and advantage distillation variants, the Cascade IR algorithm was used with the maximum number of iterations *n_c_* = 4 and the initial parity testing block length *n_block_* = 8. The cascade algorithm terminates its operation when Alice’s and Bob’s sequences become equal. The mean value of the number of iterations needed to achieve this equality is denoted by $\bar{n_{c}}$. 

The system performance was measured by the following indicators, see Table 1 and Table 2: final key length,the total length of final keys,key rate (KR),IR efficiency,the final normalized Hamming distance between Alice’s and Eve’s keys,key agreement rate (KA),leakage rate (LR), andmean block entropy.

The key rate is given by
(12)$$\mathrm{KR}=\frac{\mathrm{total}\;\mathrm{length}\;\mathrm{of}\;\mathrm{established}\;\mathrm{keys}}{\mathrm{total}\;\mathrm{length}\;\mathrm{of}\;\mathrm{input}\;\mathrm{sequences}}\cdot 100\;[\%]$$
and the information reconciliation efficiency is defined as
(13)$$\mathrm{IR}\;\mathrm{efficiency}=\frac{m}{H(A|B)}=\frac{m}{{\mathrm{nh}}_{b}{(D}_{h}\left(A,B\right))}$$
where *m* is the total number of bits exchanged over the public channel during the IR phase, *n* is the length of strings at the beginning of the IR phase, and *h_b_* is the binary entropy function defined in Equation (6). In fact, it is the relationship between the exchanged number of bits and the theoretical minimum number of bits, established in [38]. This ratio has a minimum value of 1, corresponding to an optimal IR procedure based on Slepian Wolf’s optimal source coding of correlated sources. At the same time, this quantity is a form of measure of the so-called communication complexity of the IR protocol.

The final normalized Hamming distance is given by:(14)$${\mathrm{Final}\;\mathrm{normalized}\;\mathrm{Hamming}\;(A,E)=D}_{h}\left(K_{A}{,K}_{E}\right)$$
i.e., represents the normalized Hamming distance between the final Eve’s and Alice–Bob’s keys. Ideally, these keys must be statistically independent. Then, according to Equation (8), the expected value of (14) is equal to 0.5.

The key agreement rate (KA) is calculated according to the expression
(15)$$\mathrm{KA}=\frac{{\mathrm{number}\;\mathrm{of}\;\mathrm{successful}\;\mathrm{key}\;\mathrm{establishment}\;(K}_{A}{=K}_{B})}{\mathrm{total}\;\mathrm{number}\;\mathrm{of}\;\mathrm{attempts}}\cdot 100\;[\%]$$

The leakage rate measures the amount of information per bit contained in Eve’s keys about Alice and Bob’s common keys:(16)$$\mathrm{LR}=I\left(X;Z\right){=1-h}_{b}{(D}_{h}\left(A,E\right))$$

The mean block entropy is given by
(17)$$\mathrm{Mean}\;\mathrm{block}\;\mathrm{entropy}=\frac{1}{20\;}\overset{k=20}{\underset{k=1}{\sum }}H_{k}$$
where $H_{k}$ is a block entropy of order (block size) k, given by Equation (3). This quantity measures the degree of uncertainty of the established keys. Figure 16 shows the change in $H_{k}$ for order *k* in the range from 1 to 20 for all 6 variants of the tested SKD systems.

As for the degree of randomness of the generated keys, it is common to use a selected battery of statistical tests.

Table 3 shows the results of the randomness tests of key sequences obtained by the AD and ADD protocols. The randomness tests are based on the statistical test suite developed by the US National Institute of Standards and Technology, NIST, ref. [39]. The outcome of each experiment is represented by the *p*-value as shown in Table 3. An individual test is considered to be passed successfully if the obtained *p*-value is higher than the threshold of 0.01. Following the obtained results, it was shown that the AD and ADD key sequences met the defined randomness criteria in all presented tests.

Based on all the data presented in Table 1, Table 2 and Table 3, the following conclusions can be drawn.

The SKD system based on 10-bit quantization is significantly better than the one based on 5-bit quantization. The average KR for all Eve types for the AD protocol is 4.78% for 10-bit quantization and 1.78% for 5-bit. Consequently, the 10-bit AD gives an advantage approximately 2.7 times higher than the 5-bit AD. In the category of ADD protocols, the same indicators are 9.01% for 10-bit quantization and 5.44 for 5-bit, which is an advantage of approximately 1.6 times for the 10-bit ADD.The key agreement rate (KA) is 100% for 10-bit quantization, regardless of the type of AD protocol.The quality of cryptographic keys for all tested variants of the proposed SKD system is approximately the same and meets the highest cryptographic criteria, both in terms of randomness (confirmed by the NIST test, see Table 3) and in terms of low information leakage. The expected value of the normalized Hamming distance between Eve’s and the legitimate keys is almost 0.5, see Table 2 and Table 3, indicating strong statistical independence.The price paid for the high KR obtained by the ADD protocol is an increase in communication complexity: average IR efficiency = 3.62, compared to an average value of 1.17 for the AD protocol and 10-bit quantization. Note that the IR efficiency for the AD protocol is close to unity, i.e., to the optimum value. The fact that the efficiency of the system depends little on the attacker is fascinating. It can be seen from the extremely small variations of performance indicators for all three types of Eve (EE, ME, UE), for the given quantization and AD settings. This phenomenon can be explained by the fact that the AD protocols look for parts of Alice’s and Bob’s signals which tend to be more similar to each other than to Eve’s signals. The AD protocols seem to find these parts because patterns in the primary EEG source are clearly expressed, invariant to individual variations. This mechanism also explains why asynchronous EEG signals can achieve a high KA rate.

## 5. Comparison with Related Works

Direct comparison with available studies is not possible because, so far, the EEG signal has not been used to solve the SKA problem. The dominant use of EEG in the field of security is for application as a biometric signal with the simultaneous generation of cryptographic keys, available after successful authentication [40]. Although these systems are not comparable with the proposed SKA system, primarily because the secret key distribution phase is missing, it is worth mentioning that state of the art systems of this class can generate keys of up to 192 bits with FAR/FRR parameters equal to (0.18%/0.18%), ref. [41].

Indirect comparisons can be made with those studies proposing an SKA based on a source of randomness (source model) derived from other bio-signals, whose sensors have similar functionalities to the EEG sensor. In [42,43], the Walkie-Talkie system is presented. It is a shared secret key generation scheme that allows two legitimate devices to establish a common cryptographic key by exploiting users’ walking characteristics (gait). The intuition is that the sensors on different locations on the same body experience similar accelerometer signals when the user is walking. Experimental results show that the keys generated by two independent devices on the same body can achieve up to 26 b/s, which requires approximately 5 s of walking. We should keep in mind the non-comparability of this result with the performance of our system since secret keys are established in the same physical place (the body of the subject), which is equivalent to the absence of distribution of the established secret key, similar to the previously mentioned class of biometric EEG systems.

The closest in concept to our system is the system proposed in [44], in which the source of common randomness is the ECG signal, but without the presence of an attacker (Eve). The authors present empirical results of the secret key generation speed of approximately 2 b/s, without testing the final keys for randomness.

It follows that the proposed SKA, based on asynchronous EEG signals of participants, is superior in all parameters (key generation speed, probability of successful key agreement, cryptographic quality of established keys, communication efficiency) to any published system of the same class.

## 6. Security Issues and Application

The proposed SKA system is based on the three-step SKD algorithm, for which information-theoretical security has been proven. Therefore, the key K established in this way, of length |K|= k bits, has a maximal uncertainty of H (K) = k bits, which cannot be overcome by solving associated mathematical problems, but only by the exhaustive search of all $2^{k}$ possibilities. Since it was experimentally confirmed that the value of the final normalized Hamming distance (14) is close to the ideal value of 0.5 (see Table 1 and Table 2), it is impossible to perform the so-called related-keys cryptanalytic attack, because all generated keys are uncorrelated.

The separation of the source of common randomness, and thus the SKA system from the cryptographic and telecommunication module, enables an offline procedure of generating keys at a time that suits the users. EEG signal recording can be performed in a secure environment, whose level of security depends on the situation and application scenario. It can vary from the absolute secure level (e.g., a professional Faraday cage inside of a controlled security perimeter), down to ad hoc solutions in the field. The public authenticated channel, over which the AD and IR part of the SKD protocol is performed, can be any channel (for example, the Internet) on which the participants were previously authenticated.

Since it is an asynchronous offline system, the question of the secret key agreement rate is not a critical parameter. 

Below are examples of two usage scenarios of our SKD system with a 9 b/s secret key rate:

Example 4.

Assignment. Transfer one page of printed text absolutely secretly (guaranteed information-theoretical security).

The solution. Assuming that the average printed page of text has 20,000 bits, it is necessary to apply the Vernam cypher with a one-time cryptographic key of the same length [1]. For this, we need 20000/9 = 2222 s of recorded EEG signal, which is approximately 37 min. A 37-min EEG recording session can be performed at both communication parties, at any time before the cypher text is generated and sent.

Example 5.

Assignment. Supply a pair of cryptographic devices with cryptographic keys, whose symmetric encryption algorithm has a key length of 500 bits. 

The Solution. We need 500/9 = 56 s of recorded EEG signal, which is under one minute. Therefore, it is necessary to record 1 min of EEG signal at both communication parties, any time before starting secure communication. Note that a professionally designed symmetric encryption system, whose secret key is 500 bits long, can work securely for a very long period of time in a normal mode of use, without the need to change the secret keys.

## 7. How to Increase the Secret Key Rate

In order to increase the KR, several different approaches are possible depending on the purpose of the entire cipher system.

Scenario A-Hybrid system: SKA source model + SKA channel model

After the completion of the off-line procedure for secret keys agreement with the proposed SKD system, encrypted communication on the main communication channel will begin. If an additional SKD based on the Channel model is designed (SKA_ChMod), then the equivalent KR is significantly increased, given the typical KR values for SKA_ ChMod systems (see overview of the channel models, ref. [20]). This approach would be especially effective if the main channel is wireless. The downside of this approach is the exposure of the SKA_ChMod procedure to electronic jamming, which, in critical conditions (e.g., war actions), can completely fail.

Scenario B-Change of primary EEG source

In this scenario, the SKA remains in the off-line mode of operation, retaining all the positive properties, such as robustness and high reliability. Because KR is limited by the secrecy capacity, given by (1), its increase is possible by changing the source of common randomness, for which the Ck is as large as possible. In our case, this would mean finding new transformations of the original EEG signals for which the resulting primary source would have a higher Ck. Within this approach, it is also possible to add new biometric sensors as sources of common randomness (for example ECG, gait sensors, etc.), under conditions that do not reduce the functionality of the entire system. 

Scenario C-Elimination of eavesdroppers

When Eve is independent of Alice and Bob, i.e., when Z is independent of X and Y, the secret key capacity is equal to its maximum value (2). We can view this phenomenon as a form of elimination of Eve, which potentially creates the possibility of increasing KR. 

Here is an example of a practical scenario for eliminating Eve. Imagine that the primary source for legitimate users Alice and Bob is formed on the basis of the transformation $T_{i}$, from the set of transformations T, with the property that any two elements of this set give mutually uncorrelated (orthogonal) outputs. More precisely,
(18)$$I\left(T_{i}\left(X\right);T_{i}\left(Y\right)\right)>0,\;\forall i\;I\left(T_{i}\left(X\right);T_{j}\left(Z\right)\right)=0,\;\forall j\neq i\;I\left(T_{i}\left(Y\right);T_{j}\left(Z\right)\right)=0,\;\forall j\neq i$$

If Alice and Bob choose the transformation $T_{i}$ secretly (e.g., based on previously exchanged secret keys), then Eva’s observations Z with probability (Card(T)−1)/Card(T) are independent of X and Y, where Card(T) is the number of elements in the set T. For example, if we generate transformations T by deep neural networks with millions of continuous parameters, this probability is both theoretically and practically equal to 1.

## 8. Conclusions

The paper presents a class of SKD systems whose inputs are so-called performance measures derived from asynchronously recorded EEG signals of communication parties. Experimental evaluation shows that careful selection of system parameters can give a key agreement rate KA = 100%, and a secret key rate up to KR = 9%, with good random properties, and little leakage of information (LR = 0.0003) to a potential attacker on the system. The system shows low sensitivity of performance to variations of the EEG signal of Eve (attacker), which confirms the hypothesis about the importance of synchronicity of legitimate participants, achieved by efficient AD protocols.

Our future research will be focused on reducing the communication complexity of the proposed system, and on its possible combination with other approaches for the extraction and distribution of cryptographic keys, primarily based on the so-called data exchange problem [13], as well as on the further improvement of the system in the domains of local randomness generation [45] and biometric applications [46].

## Figures and Tables

**Figure 1 entropy-23-01327-f001:**
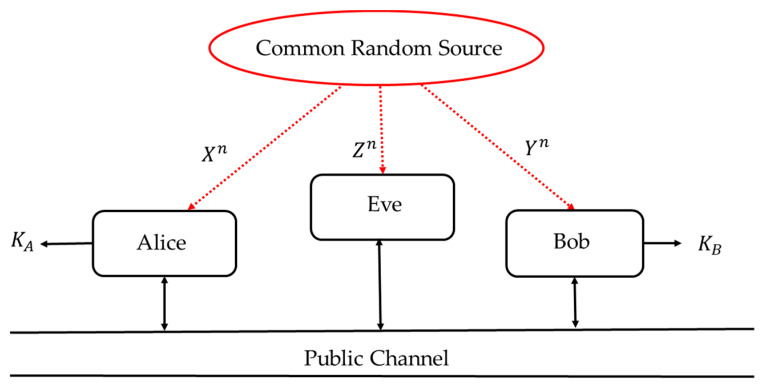
Secret-key agreement by public discussion from common information [4].

**Figure 2 entropy-23-01327-f002:**
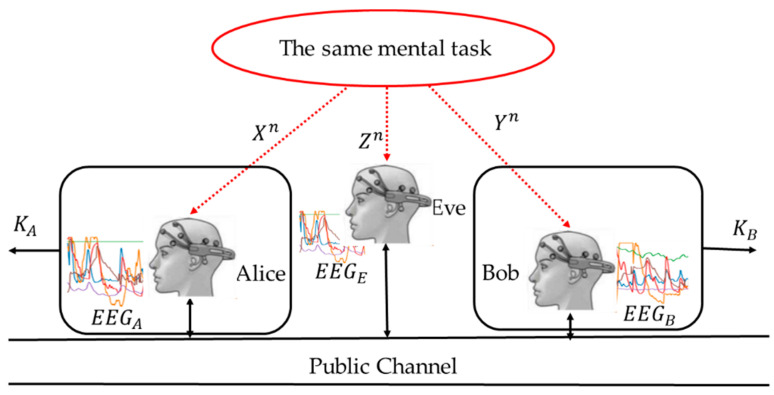
Secret-key agreement by public discussion from EEG signals asynchronously obtained during the same mental task.

**Figure 3 entropy-23-01327-f003:**
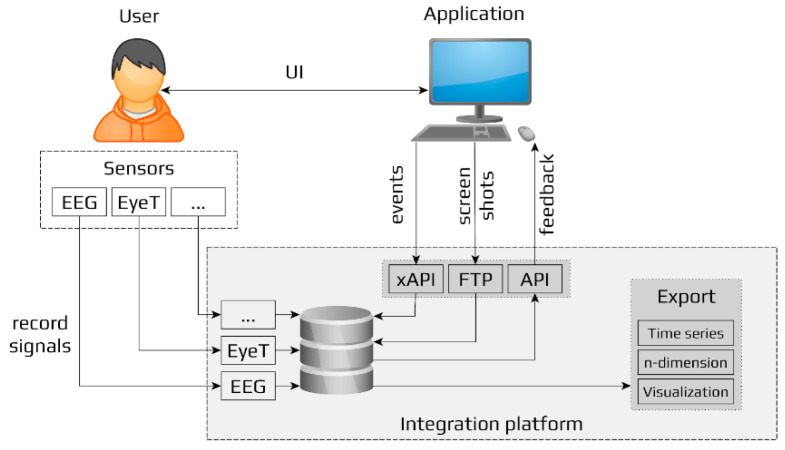
HCI monitoring and analytics platform (HCI.MAP), ref. [15].

**Figure 4 entropy-23-01327-f004:**
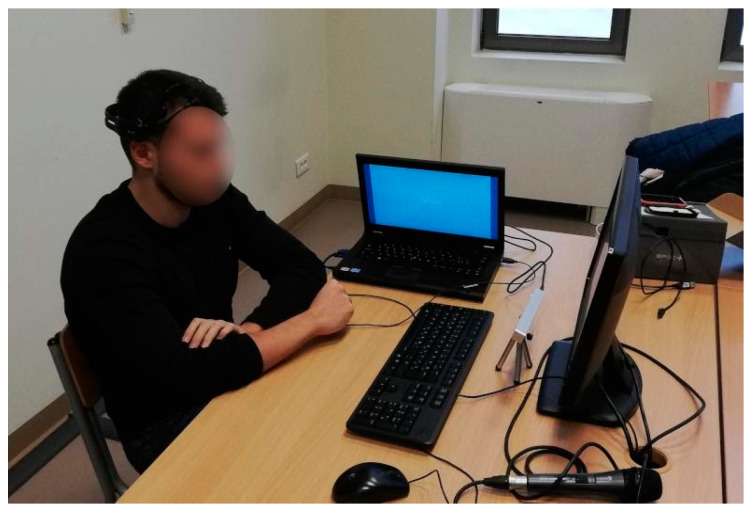
An experimental environment with EEG and eye-tracking sensors enabled, ref. [15].

**Figure 5 entropy-23-01327-f005:**
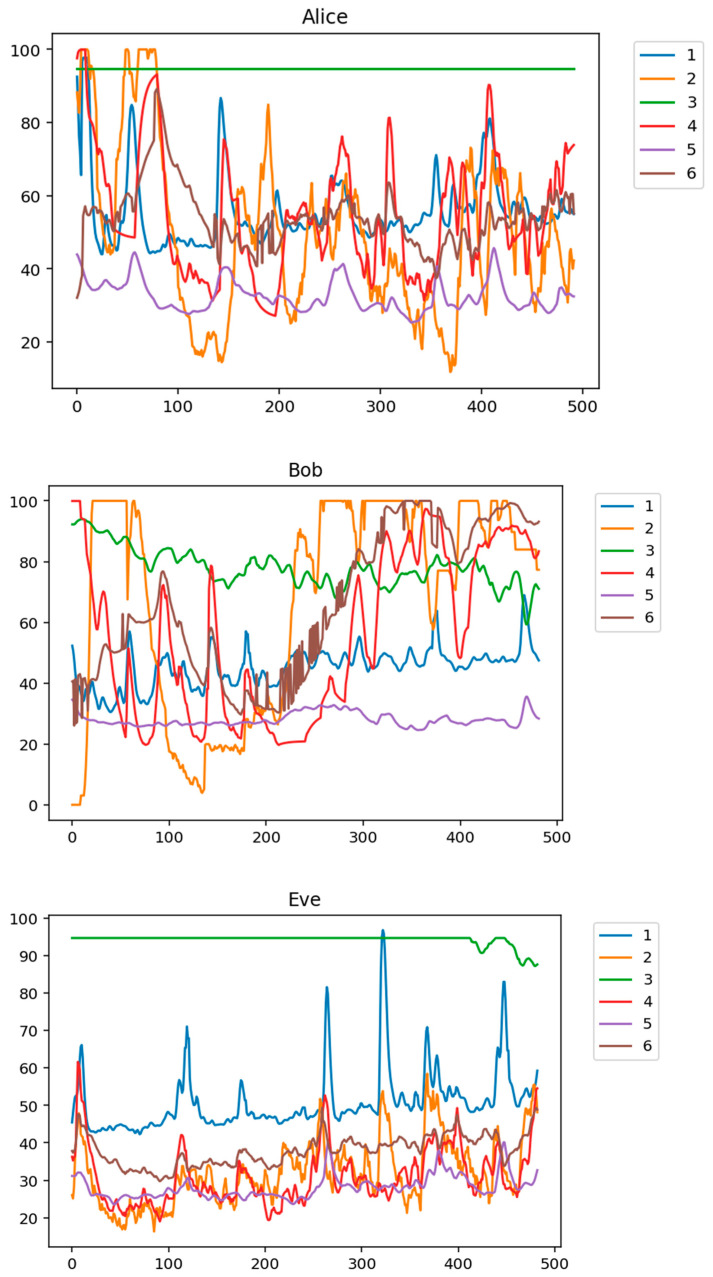
An example of performance metrics signals for Alice, Bob, and Eve, randomly selected among all 76 test participants.

**Figure 6 entropy-23-01327-f006:**
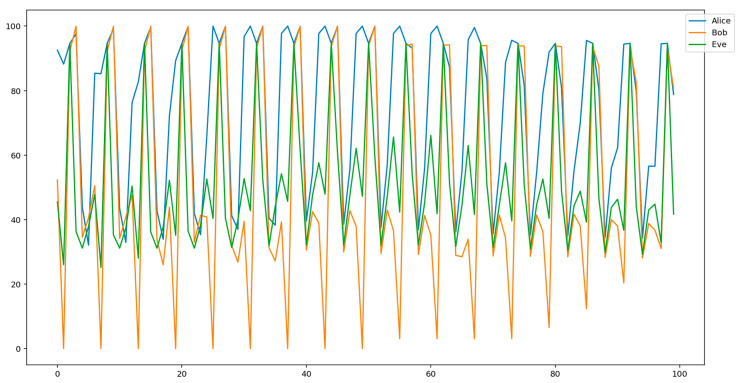
The one-dimensional signals resulting from the serialization of initially 6-dimensional signals of Alice, Bob, and Eve shown in Figure 5.

**Figure 7 entropy-23-01327-f007:**
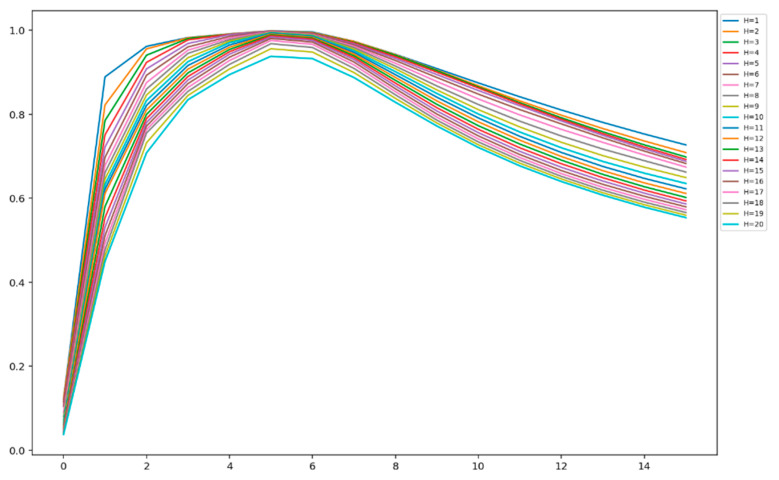
Normalized block entropies of the analyzed primary EEG source, as a function of the number of bits per sample, obtained by uniform quantization. Each curve corresponds to one of the block length values from 1 to 20.

**Figure 8 entropy-23-01327-f008:**
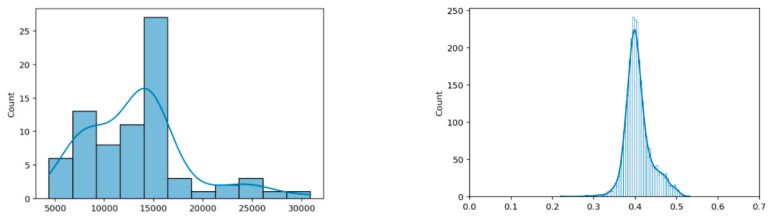
Uniform coding with 5 bits per sample (n_b_ = 5). Histogram of EEG sequence lengths for all participants (left) and histogram of normalized Hamming distances of all pairs (right): total sequence length 1,006,560 bits; mean and dispersion of normalized Hamming distances is (0.41 +/− 0.036).

**Figure 9 entropy-23-01327-f009:**
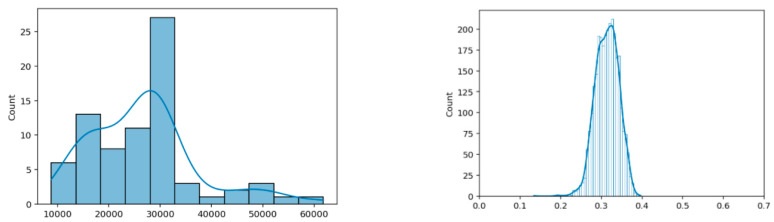
Uniform coding with 10 bits per sample (n_b_ = 10). Histogram of EEG sequence lengths for all participants (left) and histogram of normalized Hamming distances of all pairs (right): total sequence length of 2,013,120 bits; mean and dispersion of normalized Hamming distances is (0.31 +/− 0.031).

**Figure 10 entropy-23-01327-f010:**
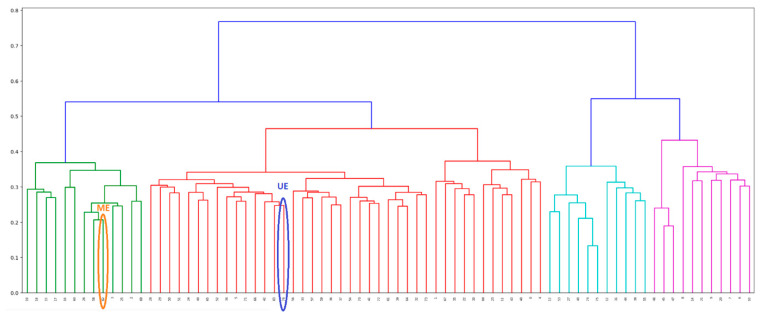
Dendrogram for hierarchical cluster analysis of the primary EEG signals of all test participants, formed by Ward’s method. “Medium evil Eve”-ME and “Uninformed Eve”-UE are marked by their acronyms, and encircled in orange and blue, respectively.

**Figure 11 entropy-23-01327-f011:**
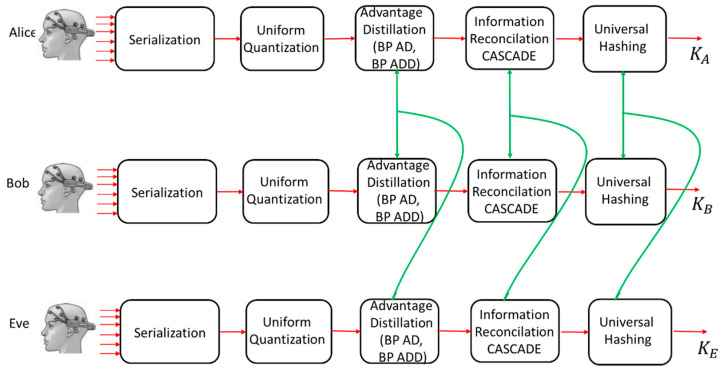
Structure of the proposed SKD system based on asynchronous EEG signals of the participants. Communications over the public channel are marked in green.

**Figure 12 entropy-23-01327-f012:**
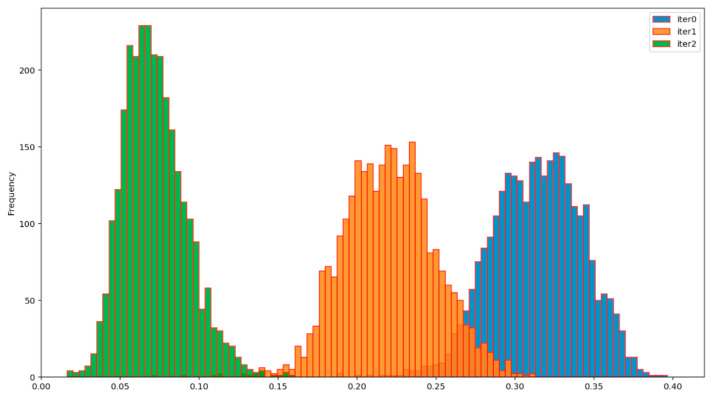
Evolution of the distribution of the normalized Hamming distances between Alice’s and Bob’s sequences during two iterations of the BP AD protocol.

**Figure 13 entropy-23-01327-f013:**
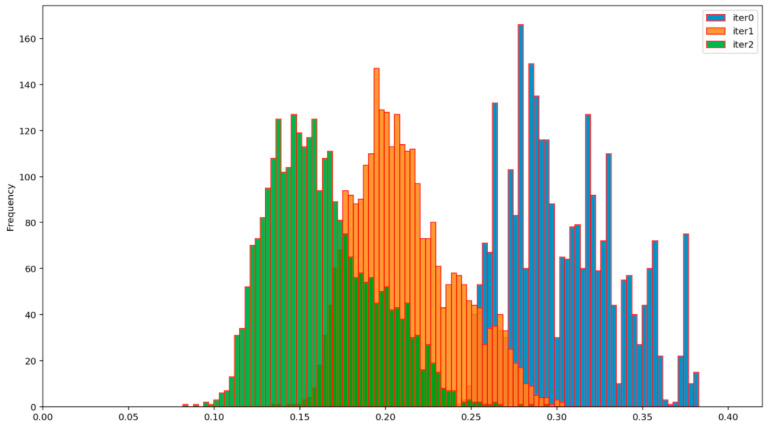
Evolution of the distribution of the normalized Hamming distances between Alice’s and Eve’s sequences during two iterations of the BP AD protocol.

**Figure 14 entropy-23-01327-f014:**
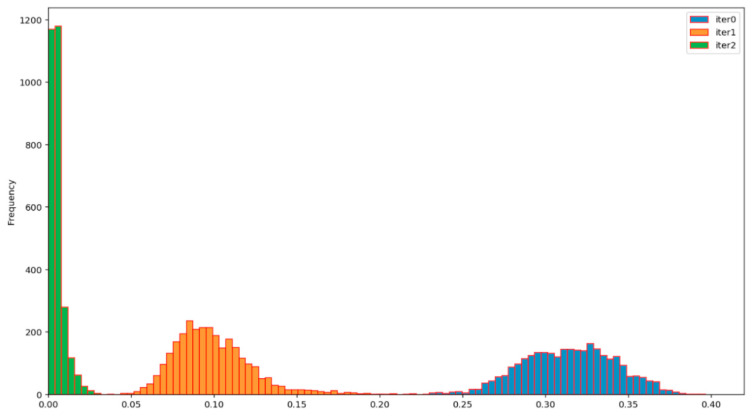
Evolution of the distribution of the normalized Hamming distances between Alice’s and Bob’s sequences during two iterations of the BP ADD protocol.

**Figure 15 entropy-23-01327-f015:**
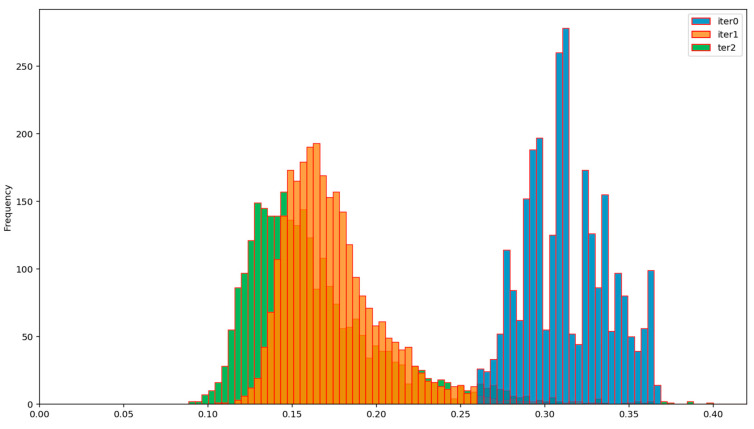
Evolution of the distribution of the normalized Hamming distances between Alice’s and Eve’s sequences during two iterations of the BP ADD protocol.

**Figure 16 entropy-23-01327-f016:**
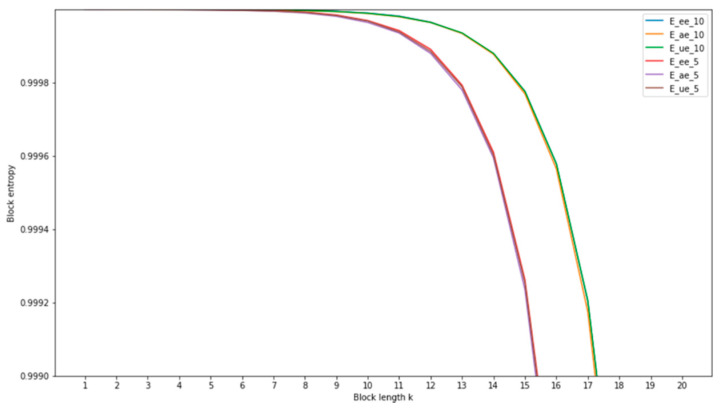
Block entropy of final keys obtained by the proposed SKD quantization system with n_b_ = 5 and with n_b_ = 10 bits for all three types of Eve (EE, ME, UE).

**Table 1 entropy-23-01327-t001:** Results for AD protocol.

Parameter	na = 2 nc = 4 nblock = 10 nb = 10			na = 2 nc = 4 nblock = 10 nb = 5		
Type of Eve	EE	AE	UE	EE	AE	UE
nc mean	2.27	2.26	2.27	2.57	2.59	2.56
Final key length (mean,std)	1301.55 ± 502.16	1290.53 ± 496.85	1301.76 ± 502.44	243.04 ± 138.77	242.48 ± 139.26	243.30 ± 137.28
Total length of final keys	3,709,416	3,581,223	3,710,007	587,184	569,341	587,576
Key rate (KR) [%]	4.79	4.75	4.79	1.79	1.77	1.79
Leakage rate	0.0006 ± 0.0010	0.0006 ± 0.0010	0.0006 ± 0.0010	0.0058 ± 0.0198	0.0053 ±0.0170	0.0057 ± 0.0168
IR efficiency	1.17 ± 0.05	1.17 ± 0.05	1.17 ± 0.05	1.17 ± 0.05	1.17 ± 0.05	1.17 ± 0.05
Final normalized Hamming (A,E)	0.4997 ± 0.0147	0.5005 ± 0.0149	0.4999 ± 0.0147	0.4997 ± 0.0527	0.5003 ± 0.0495	0.4988 ± 0.0487
Key agreement rate (KA) [%]	100	100	100	84.77	84.61	84.74
Mean block entropy (k = [1, 20])	0.9989	0.9988	0.9989	0.9926	0.9924	0.9927

**Table 2 entropy-23-01327-t002:** Results for ADD protocol.

Parameter	na = 2 nc = 4 block = 10 nb = 10			na = 2 nc = 4 block = 10 nb = 5		
Type of Eve	EE	AE	UE	EE	AE	UE
nc mean	1.37	1.37	1.37	1.92	1.93	1.92
Final key length (mean,std)	2454.28 ± 819.68	2435.94 ± 811.47	2454.09 ± 819.52	739.12 ± 297.97	743.11 ± 300.32	738.29 ± 298.51
Total length of final keys	6,994,706	6,759,745	6,994,143	2,103,530	2,059,898	2,104,116
Key rate (KR) [%]	9.04	8.96	9.04	5.44	5.44	5.44
Leakage rate (LR)	0.0003 ± 0.0005	0.0003 ± 0.0005	0.0003 ± 0.0006	0.0013 ± 0.0022	0.0012 ± 0.0024	0.0013 ± 0.0027
IR efficiency	3.63 ± 2.11	3.60 ± 2.11	3.63 ± 2.11	1.86 ± 0.63	1.85 ± 0.62	1.86 ± 0.63
Final normalized Hamming (A,E)	0.4998 ± 0.0106	0.4999 ± 0.0107	0.5005 ± 0.0109	0.5002 ± 0.0209	0.5000 ± 0.0200	0.5000 ± 0.0210
Key agreement rate (KA) [%]	100	100	100	99.86	99.89	100
Mean block entropy (k = [1, 20])	0.9994	0.9994	0.9994	0.9979	0.9979	0.9979

**Table 3 entropy-23-01327-t003:** Randomness test results of the AD and ADD key sequences. The tests are based on the statistical test suite developed by NIST, and results are presented in terms of P-values. Initial letters indicate test names: F-frequency, BF-block frequency, R-runs, LR-longest run, FFT-fast Fourier transformation, S-serial, AE-approximate entropy, CSf-cumulative sums forward, CSr-cumulative sums reverse. Both tested sequences have the same length of 12 million bits.

	F	BF	R	LR	FFT	S	AE	CSf	CSr
AD	0.9114	0.5341	0.3504	0.5341	0.9914	0.7399	0.5341	0.7399	0.0668
ADD	0.0351	0.7399	0.3504	0.0088	0.7399	0.7399	0.1223	0.7399	0.2133

## Data Availability

EEG data of all participants can be accessed from: https://github.com/hajdeger/AOP_PUB/blob/master/EEG_sve.csv (accessed on 6 October 2021).

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
