# Peer review of "Secret-Key Agreement by Asynchronous EEG over Authenticated Public Channels"

_entropy, 2021, doi:10.3390/e23101327_

Round 1

Reviewer 1 Report

The reviewer thanks the authors' efforts on improving the presentation and almost all issues are satisfactorily addressed. The reviewer would like to ask the authors to extend their response to the comment about low key rate from offline to online. In particular, a short remark about how to increase the key rate will be sufficient. Then this paper can be accepted.

Reviewer 2 Report

Response to Remark 1.

If the potential applications are cryptography and the human-computer interface, then it would be important to at least mention that ethical issues such as privacy are considered.

Response to Remarks 2 and 3.

We understand that the intended applications of the proposed method are broad. However, if the study is claiming to detect a meaningful signal from the brain, then we will need to be convinced that the system used is not just detecting noise, which is the main concern. Hence it is important for the system to be validated from the perspective of a brain specialist (neuroscientist, neurophysiologist, or neuropsychologist). For example, the labels for the plots ‘interest’, ‘engagement’, ‘excitement’, ‘stress’, ‘relaxation’, and ‘focus’ are suggesting that these mental states are detected and represented by the system. However, we need more evidence or at least an expert opinion that these are meaningful labels within the fields of neuroscience, neuropsychology, or neurophysiology. Alternatively, it may be less misleading if the labels are simply titled 1 to 6.

Reviewer 3 Report

The authors have revised their manuscript carefully, and solved the previous provious problems in this version.

On Page 2, Para. 5, Line 5, "the chapter ..." should be "the section".

Author Response

On Page 2, Para. 5, Line 5, "the chapter" ...we will change to "the section".

Round 2

Reviewer 2 Report

With respect to Remark 2, I am not convinced that what is detected reflects meaningful signal from the brain. It is unclear how the EEG would reflect brain activity during the Wisconsin card sorting test. From a neurophysiological perspective, there are still notable omissions regarding the study design. Perhaps it would be more appropriate to minimise the neuroscience focus, if there is insufficient evidence that the EEG signals are meaningful. 

Round 3

Reviewer 2 Report

The authors have addressed the majority of the issues we previously raised